# Which Factors Influence the Immensely Fluctuating CRT Implantation Rates in Europe? A Mixed Methods Approach Using Qualitative Content Analysis Based on Expert Interviews

**DOI:** 10.3390/jcm12062099

**Published:** 2023-03-07

**Authors:** Christoph Edlinger, Marwin Bannehr, Christian Georgi, David Reiners, Michael Lichtenauer, Anja Haase-Fielitz, Christian Butter

**Affiliations:** 1Department of Cardiology, Heart Center Brandenburg, 16321 Berlin, Germany; 2Faculty of Health Sciences Brandenburg, Brandenburg Medical School (MHB) “Theodor Fontane”, 16816 Neuruppin, Germany; 3Clinic of Internal Medicine II, Department of Cardiology, Paracelsus Medical University of Salzburg, 5020 Salzburg, Austria

**Keywords:** expert interview, CRT, GDP, health expenditure, disparity

## Abstract

(1) Background: Cardiac resynchronisation therapy (CRT) is nowadays an indispensable treatment option for heart failure. Although the indication is subject to clear cross-national guidelines by the European Society of Cardiology (ESC), there is immense variation in the number of implantations per 100,000 inhabitants in Europe, especially in German-speaking countries (Germany, Austria and Switzerland). The aim of the present study was to identify possible factors for these differences using a qualitative research approach. (2) Methods: Semi-standardized interviews were conducted with 11 experts in the field of CRT therapy (3 experts from Germany, 4 from Austria and 4 from Switzerland) using a pre-prepared interview template and analysed according to Mayring’s qualitative content analysis. (3) Results: The main factors identified were the costs of purchasing the devices and the financing systems of the respective healthcare systems, although cost pressure still seems to play a subordinate role in the German-speaking countries. Moreover, “lack of implementation of ESC guidelines”, “insufficient training” and “lack of medical infrastructure” could be excluded as potential reasons. (4) Conclusions: Economic factors, but not a lack of adherence to ESC guidelines, seem to have a major influence on the fluctuating implantation figures in German-speaking countries, according to the unanimous assessment of renowned experts.

## 1. Introduction

Chronic heart failure (HF), as a major cause of mortality and hospitalization, poses an immense challenge to healthcare systems in the Western world [1,2]. In patients with the appropriate indication, cardiac resynchronisation therapy (CRT) has undoubtedly established itself as a valuable non-drug therapy over the last two decades. Its safety and effectiveness have been impressively demonstrated in numerous prospective studies [3,4,5]. The indications for CRT are clearly defined in the European Society of Cardiology (ESC) guidelines. The ESC guideline on chronic heart failure (last published 2021), as well as the European Heart Rhythm Association (EHRA) guideline on cardiac pacing and cardiac resynchronization therapy, propose a CLASS IA indication for symptomatic patients in sinus rhythm with a QRS duration ≥ 150 ms and a left bundle branch block-QRS morphology and with a LVEF ≤ 35% despite optimal medical treatment [6,7]. Moreover, the European Heart Rhythm Association (EHRA) regularly publishes a report on implantation figures in the respective EHRA member countries in the form of the *EHRA White Book* [8]. Looking at the 2017 *EHRA White Book*, it is striking that there are extreme intra-European differences, even if obvious reasons, such as the socioeconomic status of the individual countries, are disregarded [9]. This is especially true for implantation rates in the German-speaking countries: Germany, Austria and Switzerland. Despite comparable socioeconomic conditions, the implantation rates per 100,000 inhabitants vary between 26.69 in Germany, 15.04 in Austria and 12.69 in Switzerland. 

A brief summary of the specific circumstances in the respective countries can be found in Table 1.

However, based on pure numbers, it seems to be possible to distinguish between overuse and underuse in different regions only to a very limited extent. While underuse may deprive patients of useful and potentially lifesaving therapy, overuse may be harmful to patients and should be avoided. Because CRT implantation requires a high level of clinical expertise, there is still some residual risk during the procedure [10]. Moreover, as with all cardiac implantable electronic devices (CIEDs), a certain percentage of patients may experience serious complications, such as device infection, during long-term treatment [11,12,13]. Not to be forgotten, it is a cost-intensive therapy, so in times of increasing resource scarcity and rising cost pressure, treating physicians are faced with the challenge of ensuring optimal allocation and selecting the “right patients”, who could have a sustainable clinical benefit [14]. 

The aim of the present study was to further analyse the differences in implantation rates within Europe using a qualitative research approach. For this purpose, German-speaking countries were selected because they have comparable socioeconomic conditions and similar medical infrastructure [9]. Semi-standardized interviews were conducted with leading experts in the field of CRT in Germany, Austria and Switzerland to identify explanatory models for the extreme differences in implantation rates within these countries.

## 2. Materials and Methods

Data Source:

The numbers of CRT implantations per 100,000 inhabitants, as well as the healthcare expenditure per 100,000 inhabitants, were calculated using data from the 2017 *EHRA White Book*, as described in detail in previous studies [9]. A graphical illustration of the strongly varying implantation figures within Europe can be found in Figure 1.

Expert Interviews:

Semi-standardized qualitative expert interviews were conducted using a pre-prepared interview guide. Participants were recruited from March 2022 to September 2022. The target group (“experts”) consisted of cardiologists with at least 5 years of clinical and scientific expertise in the field of CRT, working in a hospital in Germany, Austria or Switzerland. The aim was to obtain at least three interviews from each country to identify any country-specific differences. All participants were informed in advance by email about the study procedure. After the participants had given their consent, including audio recording and data processing, the interviews took place in person (*n* = 2), as a Zoom meeting (*n* = 4) or via telephone (*n* = 5). All interviews were conducted by the first author between May 2022 and September 2022. Interviews were tape-recorded, converted to MP3 files and later transcribed verbatim. The interviews lasted between 8 min and 35 min.

Table 2 shows the academic profile of the interviewees.

Interview Guide:

The complete interview guide can be found in Appendix A. The interview was piloted to ensure comprehensibility and feasibility. Each interviewee was first presented with the total number of CRT implantations in 2017 recorded in the *EHRA White Book*, and the calculated implantation numbers per 100,000 inhabitants for Germany, Austria and Switzerland. After an open question on how to explain these variations, the individual questions of the interview guide were worked through. If necessary, follow-up questions were asked for clarification. Furthermore, the interviewer was entitled to ask questions not listed in the interview guide if they seemed necessary to clarify the content.

Sample:

In total, 53 CRT experts were contacted in the 3 countries. Ultimately, a full interview could be conducted with 11 of these experts. Most of the participating physicians worked at university hospitals, and almost all of them were CRT implanters themselves or, alternatively, could demonstrate longstanding clinical and scientific expertise in the field of CRT.

Data Analysis:

For the data analysis, the Qualitative Content Analysis according to Mayring was carried out, whereby the collected data are divided into categories and subcategories in the form of an inductive data analysis. The expert interview with evaluation based on qualitative content analysis according to Mayring is a method that is widely used in numerous sciences, such as sociology, psychology or medical care research. A schematic illustration of the individual steps according to Mayring can be found in Figure 2. This approach makes it possible to reduce and categorise the data to enable systematic text processing. For this purpose, the transcribed interview text is first paraphrased, whereby a simplification of the text into a consistent short form is obtained by removing unnecessary passages, such as repetitions or filler words. Based on these paraphrases, further categorisation into groups and subgroups is then performed. All interviews were processed according to this procedure by the first author; to increase/validate the reliability of the coding, two randomly selected interviews were independently coded in the same way by two other authors (M.B./C.G.).

All qualitative analyses were performed using MAXQDA 2022 software (Verbi-Software, Berlin, Germany). Interview segments included in this article were translated from German into English (and, where necessary, slightly adapted for easier reading).

## 3. Results

A schematic illustration of the most relevant answers can be found in Figure 3. The original comments of the interviewed experts can be found in paraphrased form in Appendix B.

### 3.1. Adherence to the EHRA Guidelines

All experts share the opinion that the corresponding guidelines are almost fully implemented in their countries, at least as far as patients with class I indications are concerned. In this context, the Swiss experts refer to the comparatively high number of colleagues certified by the EHRA. In Austria, the experts indicated that most patients are treated in a specialized outpatient clinic for chronic heart failure, so that drug therapy options are usually fully exhausted before CRT implantation. Nevertheless, the experts in Austria and Switzerland report that there are rural/remote regions where there is a certain underuse. This is not for medical reasons in the strict sense; there are always people who have very limited mobility, even at an advanced age, or who do not go to the doctor at all.

### 3.2. Ethical Aspects

All experts strongly oppose strict age limits for device implantations. The majority believe that patients are entitled to a device, regardless of their age or insurance status. In the case of a CRT-Defibrillator implantation, the experts were reluctant, with the majority seeing a limit at around age 80. Nevertheless, almost all experts in this context state that an interindividual assessment “at eye level” by experienced physicians must always be the basis in the decision-making process. 

### 3.3. Economic Aspects

All experts are of the opinion that economic reasons might be an explanation for the immense fluctuations. Less in public hospitals than in the private sector, the experts see a certain danger of overtreatment, as it is a very well-remunerated intervention in individual regions of Europe. In direct comparison with Germany, the experts in Switzerland and Austria see a decisive difference in the hospitals’ purchasing prices. In their opinion, it is not possible for a small country to obtain similar purchase prices as a high-volume centre or a large hospital group in Germany. Nevertheless, all respondents stated that the presence of private insurance or self-paying patients, at least in public or university hospitals in their countries, had no influence on implantation practices.

### 3.4. Cost Pressure

The experts from Austria see only minor cost pressure. Only in the case of expensive technologies, such as leadless pacemakers or subcutaneous devices, would there regularly be restrictions on the part of the cost bearers. The experts in Germany are of the opinion that, despite the current crisis situation, no politician will say that there will be restrictions. In Switzerland, there would not have been any cost pressure so far. Costs have only been an issue since the Swiss-DRG system was introduced a few years ago. Further, the system is currently facing another change, and there are often enquiries from health insurers as to why one has decided on a particular therapy. Nevertheless, the experts in all three countries currently see no relevant restrictions in the treatment of their patients. In Switzerland, there seems to be a certain special role in dealing with referring physicians. Far more than in Germany and Austria, all four Swiss experts interviewed emphasize that the indication is made in strict agreement with the referring cardiologist; that in many cases each individual case is discussed with the specialist in private practice; and that in the case of elderly patients, a geriatrician is often also involved in the decision-making process. The Swiss experts also see a very rapid implementation of important “landmark studies” among colleagues in private practice, whereas the experts in Austria and Switzerland are of the opinion that it can take years until innovations are fully implemented. In the case of the Danish trial, this led to overly cautious referrals for non-ischaemic indications, according to several Swiss colleagues [15].

### 3.5. Additional Examinations

Additional examinations beyond the formal recommendations of the EHRA, such as an MRI to assess vitality, are not considered useful by the experts. Nowadays, MRI would be useful only in cases of unclear aetiology of cardiomyopathy and in ischaemic cardiomyopathy. Here, however, it is more important with regard to possible follow-up interventions, such as VT ablation or interventional valve treatment. Some experts emphasized the importance of phlebography, especially in pre-operated patients

### 3.6. Development of CRT Therapy in Recent Years

As far as the nearer past is concerned, the experts’ opinions differed strongly. While some see an increase in CRT implantation in their area, others perceive a significantly more restrained implantation behaviour. For most, an important milestone was the implementation of the quadripolar leads, which both facilitated implantation and increased the likelihood of effective biventricular pacing.

### 3.7. CRT-Pacemaker (CRT-P) vs. CRT-Defibrillator (CRT-D)

Opinions also differed considerably on the question of how to view the development of CRT-D vs. CRT-P. The majority of the experts see a strong decline in indications for CRT-D implantation, which is partly due to the results of the “Danish-trial”. However, the further development of drug therapy for heart failure has also contributed to the fact that the defibrillator component can now be dispensed with in many cases. Two experts referred in this context to the currently ongoing Reset-CRT Trial. One expert was of the opinion that the results of the Danish trial do not relieve us of the obligation to assess the patient and his or her risk on an individual basis and to make the decision on this basis. One expert stated that in the German DRG remuneration system, a CRT-D is reimbursed better than a CRT-P, and the expert sees a clear weakness of the system that could possibly also have an influence on clinical practice. One expert is even of the opinion that the “old studies”, such as the “MADIT-II trial” or the “Companion-Trial”, which are regarded as the basis for the evidence therapy of defibrillators [16,17], can no longer be used without restriction.

### 3.8. Future Prospects

The vast majority believe that CRT will continue to play an important role in clinical practice for several years, although this would depend on how rapidly conduction system pacing develops.

As far as the future is concerned, the experts see a clear trend towards leadless technology. 

In general, several experts would like to see a shift to supraregional centres, as the required “know-how” would become even greater due to the new technologies. Several experts also see complication management as a major challenge for the future, which they all agree should also take place in a supraregional centre.

## 4. Discussion

To the best of our knowledge, this is the first study to use a qualitative content analysis to address the question of why there are such fluctuating implantation numbers in Europe despite universal guidelines for all EHRA members. The German-speaking countries (Germany, Austria and Switzerland) were deliberately selected because the socioeconomic conditions and the infrastructure in terms of hospitals and physicians appear comparable. The main findings of our study can be summarized as follows:

(1) The experts of all three countries see the guidelines fully implemented in their countries. Differences seem to exist in the communication between the main hospital and the general practice sector.

(2) Economic factors, namely, the purchase price of devices and the respective reimbursement system, seem to play a role, whereas the presence of private insurance seems to play a minor part.

(3) The experts see a major challenge for the future in the increasingly sophisticated technology, as well as in complication management. Therefore, a thorough patient selection, in the sense of “eyeballing” by the experienced physician, is essential.

In Austria, too, there is now a structured “pacemaker curriculum”, although there still seems to be a lack of implementation, especially outside urban areas.

An interesting aspect is the described “self-inflicted underdiagnosis”, according to which there is a certain percentage of people in both Austria and in Switzerland who would hardly consult a cardiologist. It is difficult to verify how high this percentage is. Here, structured training for local primary care physicians might be a potential solution. As far as economic reasons for the implantation numbers are concerned, a certain trend is emerging. While the classic indication of class 1A seems to be implemented according to guidelines in all three countries, there seems to be a certain grey area for the “softer” indications.

According to the experts’ statements, both Switzerland and Austria are rather cautious in this respect. In Germany, on the other hand, people would apparently opt for implantation in the case of doubt, which is presumably reinforced by the remuneration system (DRG system) and might also have to do with the cheaper prices of the devices. In general, the experts are of the opinion that in countries where the hospital can make a profit from implantations, the guidelines might be interpreted more generously.

Unfortunately, a direct comparison is only possible to a very limited extent due to the different financing systems within Europe. This situation is further aggravated by the strongly fluctuating purchase prices of the devices. In the interest of the European community, harmonization would possibly make sense here, but this seems very difficult to imagine in view of the historically evolved structures in the individual countries. 

Given the ongoing economic challenges (COVID pandemic, Ukraine crisis), it cannot be ruled out at present that there will be a further increase in the price of devices, which could potentially have a negative impact on the implementation of the EHRA guidelines in socioeconomically weak regions.

As for the future, most interviewees continue to see a high value for CRT treatment, even though a certain paradigm shift seems to be emerging due to conduction system pacing. In any case, technical innovations and, above all, large-scale prospective studies are urgently needed to implement these innovations on a broad basis.

In general, the experts see a process of change to the effect that device therapy should increasingly be performed in supraregional high-volume centres. On the one hand, conduction pacing requires a high level of expertise, and even the follow-up examinations of devices are becoming increasingly demanding. On the other hand, the experts see a major challenge in the coming years in the careful selection of patients and even more in complication management. 

Here, standards provided by EHRA on the minimum number to be performed per year or on the European certifications of leading centres could play an important role in ultimately optimizing the quality of treatment for our patients.

## 5. Conclusions

Our qualitative content analysis with leading experts in CRT therapy in German-speaking countries showed that economic factors, namely, the purchase prices of the devices and the reimbursements by the respective healthcare system, could be the main factors for the immensely fluctuating implantation numbers of CRTs, and, in any case, a lack of adherence to the ESC guidelines does not seem to come into question.

## 6. Limitations

We see the main limitation of this study in the fact that the experts’ assessments are primarily based on common clinical practice in their respective countries. 

Nevertheless, at least some of the interviewees had professional experience, also in the neighbouring countries, or were aware of clinical practice in the neighbouring countries.

As these are high-level experts with many years of experience in the field of CRT therapy, we nevertheless consider the conclusions drawn to be valid, especially due to the unanimity of the experts in many cases.

The implantation numbers on which this paper is based are from 2017, so although the interviews reflect current practice, we cannot completely rule out the possibility that the case numbers may have changed in the meantime.

All quotations are original statements by experts, which have been minimally adapted where necessary for better comprehensibility. Individual statements on clinical practice and the political background reflect the personal views of the interviewees, and therefore, do not necessarily reflect the opinion of the authors.

## Figures and Tables

**Figure 1 jcm-12-02099-f001:**
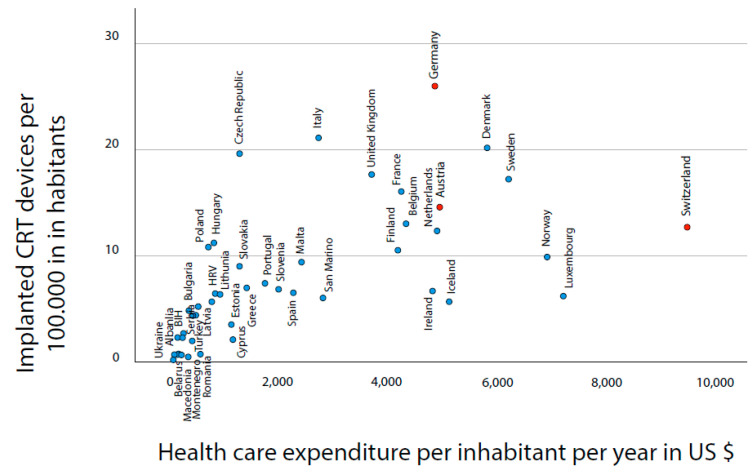
Chart for CRT implantations per 100,000 inhabitants and annual healthcare expenditure per inhabitant in Europe.

**Figure 2 jcm-12-02099-f002:**
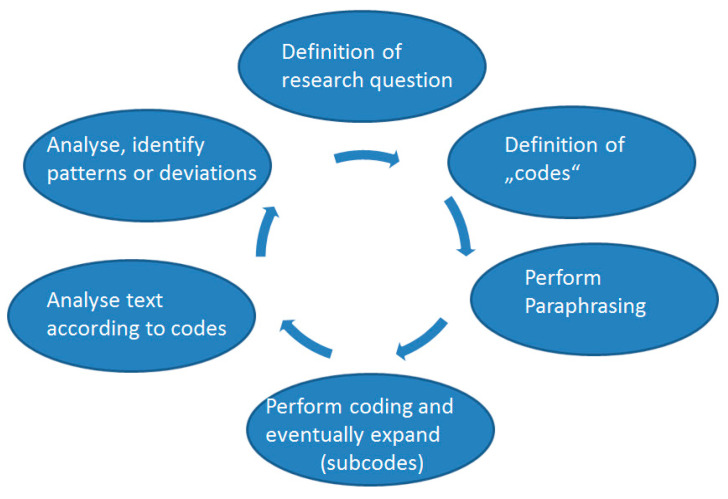
The method according to Mayring: Schematic presentation of the working steps.

**Figure 3 jcm-12-02099-f003:**
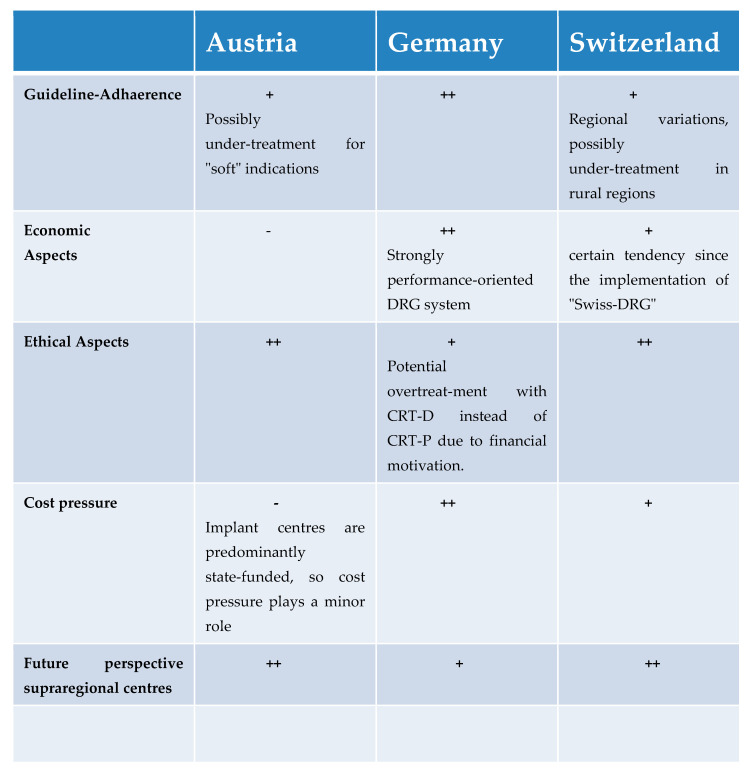
Schematic illustration of the most relevant answers. + = applies. ++ = strongly applies. - = applies less.

**Table 1 jcm-12-02099-t001:** Background information on the respective countries.

	Austria	Germany	Switzerland
CRT Implantations absolute/per 100,000 Inhabitants	1270/14.58	27,000/26.69	1038/12.69
Health Expenditure (percent of GDP)	11.2%	11.3%	11.7%
Hospitals (per 100,000 Inhabitants)	3.3	3.9	3.5
Hospital beds (per 100,000 Inhabitants)	758.6	822.8	457.2
Remuneration System	LKF-System	DRG-System	DRG-Swiss
Country-specific Circumstances	In the LKF system, a predefined number of interventions results in a cut-off, so that in the end, less is earned per additional intervention.	The DRG system is performance-oriented; accordingly, an increase in the number of cases can lead to a financial benefit.	The Swiss DRG is inspired by the German system, which changed reimbursement from a fee-for-service per diem rate to a fixed rate per case
	As most CRT implantations take place exclusively in supraregional specialised clinics, there is inevitably a certain urban/rural divide.	There are large hospital associations, so that favourable purchase prices can be obtained through high unit numbers.	In comparison with neighbouring countries, the devices are considered to be disproportionately cost-intensive in terms of purchase prices.

**Table 2 jcm-12-02099-t002:** Profile of the interviewed experts.

Overall	*n* = 11
Germany	*n* = 3
Austria	*n* = 4
Switzerland	*n* = 4
University hospital	*n* = 8
Female/Male	*n* = 2/*n* = 9
Experience in CRT treatment	
3–5 years	*n* = 2
5–10 years	*n* = 2
>10 years	*n* = 7

## Data Availability

The data presented in this study are available on request from the corresponding author.

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
