# Peer review of "Which Factors Influence the Immensely Fluctuating CRT Implantation Rates in Europe? A Mixed Methods Approach Using Qualitative Content Analysis Based on Expert Interviews"

_jcm, 2023, doi:10.3390/jcm12062099_

Round 1

Reviewer 1 Report

The data from the EHRA registry are from 2017. In the last 5 years, there has been a big change in rates of implantation, and the modality of devices implanted (conduction system pacing, CRT ...) so in my opinion the reference data are outdated. Unfortunately, despite that the EHRA registry was very useful the data has stopped being collected since the mentioned year so we don't have new data.  

Also, to draw adequate conclusions from semi-standardized interviews too few experts were contacted.

Author Response

We would like to thank you for the comments as well as the constructive criticism. Please find our statements attached.

Reviewer 2 Report

Thank you for this very interesting manuscript exploring the differences between three European countries with regards to the CRT implantation strategies and the framework in which these decisions are made.

I a minor remarks which would lead to a broader insight into the reality of HFrEF patients in these countries:

1. You describe descrepancies between countries with regards to how man CRT devices are implanted per 100,000 people. Do the discrepancies also exist with regards to medication? Maybe patients in Austria/Switzerland are prescribed with more guideline adherent treatment strategies. 

2. How does the prognosis in these patients differ across countries?

3. Is there a difference in the structure who sees the patient in an outpatient setting on a routine basis prior to a potential implantation / after an implantation? 

Author Response

(The authors gave the same response as above.)

Reviewer 3 Report

Though the scientific value of the paper is moderate, I read it with interest because it perfectly describes the problems related to CRT therapy in the year 2023. The experts' opinions are very valuable and reflects the concerns of all cardiologists. It'a a pity that the number of interrogated experts is so small and is limited to German spoken countries only. However, in spite of the above-mentioned reservations, I support the publication of the report in a section dedicated to a relatively narrow group of experts in that area.

Author Response

(The authors gave the same response as above.)

Reviewer 4 Report

The content is well written and presents points of interest for the JCM readers. 

It results that beyond guidelines indications, economic availability has a major role. We can imagine how discriminating applications may exist for poor populations!

I think this is a point of moral reflection on human dignity and rights. Of course, the report doesn't contain technical or scientific novelties.

Author Response

(The authors gave the same response as above.)
